# FAST RoPE ATTENTION: COMBINING THE POLYNOMIAL METHOD AND FAST FOURIER TRANSFORM

## ABSTRACT

The transformer architecture has been widely applied to many machine learning tasks. A main bottleneck in the time to perform transformer computations is a task called attention computation. [Alman and Song, NeurIPS 2023] have shown that in the bounded entry regime, there is an almost linear time algorithm to approximate the attention computation. They also proved that the bounded entry assumption is necessary for a fast algorithm assuming the popular Strong Exponential Time Hypothesis.

A new version of transformer which uses position embeddings has recently been very successful. At a high level, position embedding enables the model to capture the correlations between tokens while taking into account their position in the sequence. Perhaps the most popular and effective version is Rotary Position Embedding (RoPE), which was proposed by [Su, Lu, Pan, Murtadha, Wen, and Liu, Neurocomputing 2024].

A main downside of RoPE is that it complicates the attention computation problem, so that previous techniques for designing almost linear time algorithms no longer seem to work. In this paper, we show how to overcome this issue, and give a new algorithm to compute the RoPE attention in almost linear time in the bounded entry regime. (Again, known lower bounds imply that bounded entries are necessary.) Our new algorithm combines two techniques in a novel way: the polynomial method, which was used in prior fast attention algorithms, and the Fast Fourier Transform.

## 1 INTRODUCTION

Large language models (LLMs) are among the most impactful tools in modern machine learning. LLMs such as Transformer (Vaswani et al., 2017a), BERT (Devlin et al., 2018), GPT-3 (Brown et al., 2020), PaLM (Chowdhery et al., 2022), OPT (Zhang et al., 2022), GPT-4 (Achiam et al., 2023), Gemini (Team et al., 2023), Gemini 1.5 (Reid et al., 2024), Claude3 (Anthropic, 2024), GPT-4o (OpenAI, 2024a), o1 (OpenAI, 2024b), can process natural language more effectively than smaller models or traditional algorithms. This means that they can understand and generate more complex and nuanced language, which can be useful for a variety of tasks such as language translation, question answering, and sentiment analysis. LLMs can also be adapted to multiple purposes without needing to be retained from scratch.

**Attention Computation.** LLMs currently require massive time and computing resources to perform at scale. The major bottleneck to speeding up LLM operations is the time to perform a certain operation called an attention matrix computation (Vaswani et al., 2017a; Radford et al., 2018; Devlin et al., 2018; Radford et al., 2019; Brown et al., 2020; Wang et al., 2020; Kitaev et al., 2020). These computations ask us to multiply the attention matrix $A$ with another value token matrix $V \in \mathbb{R}^{n \times d}$. More precisely, given three matrices $Q, K, V \in \mathbb{R}^{n \times d}$ (the query, key, and value token matrices), the goal is to output (an approximation of) the $n \times d$ matrix $\mathsf{Att}(Q, K, V)$ defined by $\mathsf{Att}(Q, K, V) := D^{-1}AV$ where the attention matrix $A \in \mathbb{R}^{n \times n}$ and diagonal matrix $D \in \mathbb{R}^{n \times n}$ are defined as $A := \exp(QK^\top/d)$ (with $\exp$ applied entry-wise), and $D := \mathrm{diag}(A\mathbf{1}_n)$. Here, $n$ is the input sequence length, and $d$ is the embedding dimension of the model, and one typically considers $d \ll n$ like $d = \Theta(\log n)$ in the time-intensive case of modeling long sequences.

The straightforward algorithm for this problem runs in roughly quadratic time. Moreover, there are known complexity-theoretic lower bounds (Keles et al., 2023; Alman & Song, 2023) proving that the problem cannot be solved in truly subquadratic time in the case when the input matrices $Q, K, V$ have large entries, assuming a popular conjecture from fine-grained complexity theory called the Strong Exponential Time Hypothesis (SETH Impagliazzo & Paturi (2001)) which we discuss more shortly.

In order to circumvent this lower bound, and inspired by the fact that the entries of the input matrices are typically bounded in realistic inputs (Zafrir et al., 2019; Katharopoulos et al., 2020b), a recent faster, almost linear-time algorithm (Alman & Song, 2023) was given, assuming that $\|Q\|_\infty, \|K\|_\infty, \|V\|_\infty$ are all bounded. Here the $\ell_\infty$-norm denotes that $\|Q\|_\infty := \max_{i,j} |Q_{i,j}|$. Rather than explicitly compute all the entries of the attention matrix $A$, Alman & Song (2023) only *implicitly* uses it, by using an algorithmic tool called the *polynomial method*.

More precisely, they present two results, showing that when $d = O(\log n)$, there is a sharp transition in the difficulty of attention computation at $B = \Theta(\sqrt{\log n})$. First, if $B = o(\sqrt{\log n})$, then there is an $n^{1+o(1)}$ time algorithm to approximate $\mathrm{Att}(Q, K, V)$ up to $1/\mathrm{poly}(n)$ additive error. Second, if $B = \Theta(\sqrt{\log n})$, then assuming SETH, it is impossible to approximate $\mathrm{Att}(Q, K, V)$ up to $1/\mathrm{poly}(n)$ additive error in truly subquadratic time $n^{2-\Omega(1)}$. In other words, if $B = o(\sqrt{\log n})$, then the polynomial method gives an almost linear-time algorithm, and if $B$ is any bigger, then it is *impossible* to design an algorithm that substantially improves on the trivial quadratic time algorithm, no matter what algorithmic techniques one uses.

**Bounded entries in practice.** The theoretical results of Alman & Song (2023) offer an explanation for a phenomenon commonly observed in practice: attention computation becomes significantly more efficient when the input matrices have smaller entries. Previous work on LLM implementations has noted similar observations; algorithmic techniques like quantization (Zafrir et al., 2019) and low-degree polynomial approximation (Katharopoulos et al., 2020b), which result in bounded or low-precision entries, can dramatically accelerate LLM operations. See, for example, the discussions in (Zafrir et al., 2019, Section 2) and (Katharopoulos et al., 2020b, Section 3.2.1).

**RoPE: Rotary Position Embedding.** In this paper, we study a variant on attention called RoPE attention. At a high level, RoPE gives more expressive power to the model in exchange for making the computational problem more complicated. In particular, many prior algorithms, such as the algorithm of Alman & Song (2023), no longer apply to RoPE, for fundamental reasons we will discuss.

RoPE was proposed by Su et al. (2024) and has been used extensively in large-scale industrial models. Examples which are known to use RoPE include the open-source models released by Meta such as Llama (Touvron et al. (2023a), see page 3), Llama 2 (Touvron et al. (2023b), see page 5), Llama 3 (AI (2024) and page 7 of Llama Team (2024)), and the close-source LLM Claude 3 (Anthropic (2024)) released by Anthropic. Apple also incorporates RoPE into their LLM architecture (see McKinzie et al. (2024) and page 3 of Gunter et al. (2024)).

The idea behind RoPE is to rotate the query and key vectors in the self-attention mechanism. The rotation is position-dependent and designed such that the inner product between any two position-encoded vectors reflects their relative positions. Intuitively, the $R_{j-i}$ matrices we define below will rotate embedding vectors according to their position in the input, so that in the RoPE attention mechanism, pairs of tokens with a longer relative distance will have smaller correlation.

We now briefly describe the mathematical definition of the RoPE method. We will make use of $2 \times 2$ rotation matrices, which for an angle of rotation $\theta$, can be written as

$$R(\theta) := \begin{bmatrix} \cos\theta & -\sin\theta \\ \sin\theta & \cos\theta \end{bmatrix}.$$

As above, we let $n$ be the input sequence length, and $d$ the embedding dimension. We assume here that $d$ is even.

For $i, j \in [n]$, we now define the overall relative rotation matrix for tokens at positions $j$ and $i$, which we denote by $R_{j-i} \in \mathbb{R}^{d \times d}$. As indicated by the notation, it depends only on the difference $j - i$.

$R_{j-i}$ is defined as a diagonal block matrix with $d/2$ blocks of size $2 \times 2$ along the diagonal, given by

$$R_{j-i} = \begin{bmatrix} R((j-i)\theta_1) & 0 & \cdots & 0 \\ 0 & R((j-i)\theta_2) & \cdots & 0 \\ \vdots & \vdots & \ddots & \vdots \\ 0 & 0 & \cdots & R((j-i)\theta_{d/2}) \end{bmatrix}.$$

The angle frequencies are given by $\theta_k = \alpha^{-2(k-1)/d}$ for $k \in [d/2]$. Here one thinks of the angle $\alpha$ as a fixed constant for all $i$ and $j$; in the original RoPE it is about $10^4$ (see details in Equation (15) in page 5 of Su et al. (2024)).

These $R_{j-i}$ matrices are incorporated into attention as follows. Let $W_Q, W_K, W_V \in \mathbb{R}^{d \times d}$ denote the model weights. Let $X \in \mathbb{R}^{n \times d}$ denote the representation of length-$n$ sentence. Then, we define the new attention matrix $A \in \mathbb{R}^{n \times n}$ by, for $i, j \in [n]$,

$$A_{i,j} := \exp(\underbrace{X_{i,*}}_{1 \times d} \underbrace{W_Q}_{d \times d} \underbrace{R_{j-i}}_{d \times d} \underbrace{W_K^\top}_{d \times d} \underbrace{X_{j,*}^\top}_{d \times 1}). \tag{1}$$

As in the usual attention mechanism, the final goal is to output an $n \times d$ size matrix $D^{-1}AXW_V$ where $D := \mathrm{diag}(A\mathbf{1}_n) \in \mathbb{R}^{n \times n}$.

**Formulation of RoPE Attention.** In this paper, we give a new algorithm for RoPE attention. We now formally define the problem we will solve. Notably, our algorithm actually solves the following *generalization* of RoPE attention, which captures RoPE (as we described it above) as well as many natural variants on RoPE that future work may want to consider. We emphasize that changing the many parameters which go into the RoPE definition would still be captured by our generalization below.

**Definition 1.1** (A General Approximate RoPE Attention Computation, ARAttC). *Let $B > 0$ and $\epsilon > 0$ denote two parameters. Given a set of matrices $W_{-(n-1)}, \cdots W_{-1}, W_0, W_1, \cdots W_{n-1} \in \mathbb{R}^{d \times d}$ where $\mathrm{supp}(W_i) \subset S$ for all $i \in \{-(n-1), \cdots, -1, 0, 1, \cdots, n-1\}$. Here $S \subseteq [d] \times [d]$ and $|S| = O(d)$. Given $Q \in \mathbb{R}^{n \times d}$, $K \in \mathbb{R}^{n \times d}$, and $V \in \mathbb{R}^{n \times d}$ with the guarantee that $\|Q\|_\infty, \|K\|_\infty, \|V\|_\infty \leq B$ and $\|W\|_\infty \leq 1$. We define matrix $A \in \mathbb{R}^{n \times n}$ as, for $i, j \in [n]$,*

$$A_{i,j} := \exp(\underbrace{Q_{i,*}}_{1 \times d} \underbrace{W_{i-j}}_{d \times d} \underbrace{K_{j,*}^\top}_{d \times 1}/d), \forall i \in [n], j \in [n]$$

*We define $D := \mathrm{diag}(A\mathbf{1}_n)$. The goal of Rotated attention computation is to output a matrix $T \in \mathbb{R}^{n \times d}$ such that $\|T - \mathsf{ARAttC}\|_\infty \leq \epsilon$ is small, where $\mathsf{ARAttC} := D^{-1}AV$. For matrix $M$, we use $\|M\|_\infty := \max_{i,j} |M_{i,j}|$. Note that the $1/d$ factor inside $\exp$ in the definition of $A$ is a normalization factor.*

**Remark 1.2.** *RoPE attention as defined above (Eq. (1)) corresponds to this problem where we restrict each of the matrices $W_i \in \mathbb{R}^{d \times d}$ for all $i \in \{-(n-1), , \cdots, -1, 0, 1, \cdots, n-1\}$ in Definition 1.1 to be diagonal block matrices, where each matrix has $d/2$ blocks and each block has size $2 \times 2$.*

**Our Results.**

Our main result is a new algorithm which computes General Approximate RoPE Attention Computation in almost linear time:

**Theorem 1.3** (main result, upper bound). *Suppose $d = O(\log n)$ and $B = o(\sqrt{\log n})$. There is an $n^{1+o(1)}$ time algorithm to approximate $\mathsf{ARAttC}$ up to $\epsilon = 1/\mathrm{poly}(n)$ additive error.*

In other words, although RoPE attention is more complicated than the usual attention, we are able to achieve the same running time for this more expressive version. This is, to our knowledge, the first fast algorithm for RoPE attention with provable guarantees. As we will discuss more shortly, there is a substantial barrier to using prior algorithmic techniques for attention in the setting of RoPE attention, and we overcome this barrier using a novel approach combining the polynomial method with Fast Fourier transforms.

Furthermore, we prove that the bound of $B = o(\sqrt{\log n})$ used by our algorithm is necessary, since when $B$ is any bigger, it is impossible to design a truly subquadratic time algorithm:

**Theorem 1.4** (main result, lower bound). *Assuming* SETH, *for every* $q > 0$, *there are constants* $C, C_a, C_b > 0$ *such that: there is no* $O(n^{2-q})$ *time algorithm for the problem* ARAttC$(n, d = C \log n, B = C_b \sqrt{\log n}, \epsilon = n^{-C_a})$.

To emphasize, our Theorem 1.4 doesn't just prove that our algorithmic approach cannot give a nontrivial algorithm when $B = \Omega(\sqrt{\log n})$, but more generally that it is impossible to design a nontrivial algorithm, no matter what algorithmic techniques one uses.

**Technique Overview: Limitation of Prior Techniques**

Prior fast algorithms with provable guarantees for attention are critically based on an algorithmic technique called the *polynomial method* (Alman & Song, 2023; 2024a;b). This is a technique for finding low-rank approximations of certain structured matrices. More precisely, suppose $M \in \mathbb{R}^{n \times n}$ is a low-rank matrix, and $f : \mathbb{R} \to \mathbb{R}$ is any function. Let $f(M)$ denote the matrix where $f$ is applied entry-wise to $M$. In general, although $M$ is low-rank, the matrix $f(M)$ may be a full-rank matrix. However, the polynomial method says that if $f$ can be approximated well by a low-degree polynomial, then $f(M)$ can be approximated well by a low-rank matrix. Since the usual attention matrix is defined by applying $\exp$ entry-wise to a low-rank matrix, prior algorithms approximate $\exp$ with a polynomial, then uses the polynomial method to approximate the attention matrix with a low-rank matrix which can be used to quickly perform the necessary linear-algebraic operations.

Although this approach has been successful in prior work on designing faster algorithms for many problems related to attention, it fundamentally cannot apply to RoPE attention. The key issue is that in RoPE attention, the underlying matrix which $\exp$ is applied to no longer needs to have low rank. Indeed, let $A$ denote the RoPE attention matrix (defined in Equation (1) above) and let $M$ denote $A$ before it was entry-wise exponentiated. Even in the simplest case $d = 1$, one can see that by picking the $R_{j-i}$ entries appropriately, one can choose $M$ to be *any* circulant matrix (i.e., matrix whose $(i, j)$ entry depends only in $j - i$). The polynomial method then cannot be used to argue that $A$ is approximately low-rank, since $M$ itself is not low-rank.

**Technique Overview: Combining the Polynomial Method and Fast Fourier Transform**

Although circulant matrices are typically not low-rank matrices, there is a vast literature on algorithms for manipulating them using the Fast Fourier transform. Notably, it is not hard to notice that applying *any* function entry-wise to a circulant matrix results in another circulant matrix, so if $M$ were indeed a circulant matrix as described in the previous paragraph, one could use the Fast Fourier transform to perform operations with the resulting matrix $A$.

However, even in the case of $d = 1$, the matrix $M$ can actually be a more general type of matrix which we call a *rescaled circulant matrix*. This is a matrix of the form $D_1 C D_2$ for diagonal matrices $D_1, D_2$ and circulant matrix $C$. Unfortunately, applying a function entry-wise to a rescaled circulant matrix need not result in another rescaled circulant matrix.

Our main algorithmic idea is a new version of the polynomial method: we prove that if $M$ is a rescaled circulant matrix, or even a sum of a small number of rescaled circulant matrices, and one applies a function $f$ entry-wise to $M$ such that $f$ has a low-degree polynomial approximation, then the resulting matrix can be approximated by a sum of a relatively small number of rescaled circulant matrices. In our case, we use this to write the RoPE attention matrix as a sum of rescaled circulant matrices, each of which is then manipulated using the Fast Fourier transform to yield our final algorithm.

We believe our new approach, of applying polynomial approximations entry-wise to structured matrices other than low-rank matrices, may be broadly applied in other settings as well. Although the polynomial method has been applied in many algorithmic contexts, to our knowledge, a version of the polynomial method like this has not been used before.

**Algorithmic techniques in practice.** We emphasize that our two core techniques, the polynomial method and Fast Fourier transform, are both prevalent in practice. The polynomial method is particularly used in numerous practical algorithms for attention (Banerjee et al., 2020; Keles et al., 2023; Zhang et al., 2024). For example, see detailed discussions in (Zhang et al., 2024, Section 4.1). Our new algorithm improves on these approaches in part by using *theoretically optimal* polynomials for exponentials, and combining them with the Fast Fourier transform, to give provable guarantees

about their correctness and near linear running time. To our knowledge, the Fast Fourier transform has not been used in this way in prior attention algorithms.

**Roadmap.** In Section 2, we present our related work. In Section 3, we define certain basic notations for linear algebra. In Section 4, we start with solving the linear case. In Section 5, we explain how to handle the exp units. In Section 6, we provide the hardness result. Finally, we provide a conclusion in Section 7.

## 2 RELATED WORK

**Polynomial Method for Attention.**

Alman & Song (2023; 2024b) utilize polynomial kernel approximation techniques proposed by Aggarwal & Alman (2022) to speed up both training and inference of a single attention layer, achieving almost linear time complexity. This method is further applied to multi-layer transformer (Liang et al., 2024c), tensor attention (Alman & Song, 2024a; Liang et al., 2024e), LoRA (Hu et al., 2024b), Hopfield model (Hu et al., 2024a), differentially private cross attention (Liang et al., 2024d), and Diffusion Transformer (Hu et al., 2024c). We will also use the polynomials of (Aggarwal & Alman, 2022) here.

**Other Algorithms for Computing Attention.**

Due to its quadratic time complexity with respect to context length (Vaswani et al., 2017b), the attention mechanism has faced criticism. To address this issue, various approaches have been employed to reduce computational overhead and improve scalability, including sparse attention (Child et al., 2019; Beltagy et al., 2020; Zaheer et al., 2020; Hubara et al., 2021; Kurtic et al., 2023; Frantar & Alistarh, 2023; Shi et al., 2023a; Li et al., 2024b; Han et al., 2024; Liang et al., 2024a), low-rank approximations (Razenshteyn et al., 2016; Li et al., 2016; Hu et al., 2022; Zeng & Lee, 2024; Hu et al., 2024b), and kernel-based methods (Charikar et al., 2020; Liu & Zenke, 2020; Deng et al., 2023a; Zandieh et al., 2023; Liang et al., 2024b). Additionally, linear attention has emerged as a significant fast alternative to softmax attention, prompting substantial research in this area (Tsai et al., 2019; Katharopoulos et al., 2020a; Schlag et al., 2021; Zhang et al., 2023; Sun et al., 2023; Ahn et al., 2024; Shi et al., 2023b; Zhang et al., 2024; Deng et al., 2023b; Li et al., 2024a).

**Fast Fourier transform.**

The Fast Fourier transform algorithm (Cooley & Tukey, 1965) can multiply the $n$ by $n$ Discrete Fourier transform matrix times an input vector in $O(n \log n)$ time. This algorithm is impactful in many areas, including image processing, audio processing, telecommunications, seismology, and polynomial multiplication. There has been much modern reseaqrch focused on further speeding up the Fast Fourier transform, including by decreasing the number of needed arithmetic operations (Sergeev, 2017; Alman & Rao, 2023), reducing the sample complexity in the sparse setting (Candes & Tao, 2006; Rudelson & Vershynin, 2008; Blumensath & Davies, 2010; Needell & Vershynin, 2010; Bourgain, 2014; Haviv & Regev, 2017; Nakos et al., 2019), and reducing the running time in the sparse setting (Gilbert et al., 2012; Hassanieh et al., 2012a;b; Indyk & Kapralov, 2014; Indyk et al., 2014; Price & Song, 2015; Moitra, 2015; Kapralov, 2016; 2017; Chen & Price, 2019b;a; Kapralov et al., 2019; Jin et al., 2023; Song et al., 2023). These algorithmic advances can be directly applied to compute the Fourier transforms which arise in our algorithm below.

## 3 PRELIMINARIES

In Section 3.1, we define several notations. We discuss some backgrounds for fast circulant transform. In Section 3.2, we provide a tool from previous work about how to control error by using low-degree polynomial to approximate exponential function. In Section 3.3, we discuss some backgrounds about fast circulant transform. In Section 3.4, we define rescaled circulant matrix and provide some basic tools for it.

## 3.1 Notation

For nonnegative integer $n$, we use $[n]$ to denote set $\{1, 2, \cdots, n\}$. For a vector $a$, we use $\mathrm{diag}(a)$ to denote the diagonal matrix where the $(i,i)$-th entry is $a_i$. For a matrix, we use $\mathrm{supp}$ to denote the support of the matrix, i.e., the set of entries where the matrix is nonzero. For a matrix $A$, we use $A^\top$ to denote transpose of $A$. Given two vectors $a, b$ of the same length, we use $a \circ b$ to denote their entry-wise product, i.e., the vector where the $i$-th entry is $a_i b_i$. Given two matrices $A, B$ of the same dimensions, we similarly use $A \circ B$ to denote their entry-wise Hadamard product, i.e., the matrix where the $(i,j)$-th entry is $A_{i,j} B_{i,j}$. For a matrix $A$ and a non-negative integer $t$, we use $A^{\circ t} := \underbrace{A \circ A \circ \cdots \circ A}_{t \text{ terms}}$, i.e., $(A^{\circ t})_{i,j} = A_{i,j}^t$.

## 3.2 Polynomial Approximation of Exponential

Here, we will explain a technical tool for controlling the error dependence of our approximate algorithm. In particular, will use the following optimal-degree polynomial approximation of the exponential function.

**Lemma 3.1** (Aggarwal & Alman (2022)). *Let $B > 1$ and let $\epsilon \in (0, 0.1)$. There is a polynomial $P : \mathbb{R} \to \mathbb{R}$ of degree $g := \Theta\left(\max\left\{\frac{\log(1/\epsilon)}{\log(\log(1/\epsilon)/B)}, B\right\}\right)$ such that for all $x \in [0, B]$, we have*

$$|P(x) - \exp(x)| < \epsilon.$$

*Furthermore, $P$ can be computed efficiently: its coefficients are rational numbers with $\mathrm{poly}(g)$-bit integer numerators and denominators which can be computed in $\mathrm{poly}(g)$ time.*

## 3.3 Fast Circulant Transform

**Definition 3.2** (Circulant matrix). *Let $a \in \mathbb{R}^n$ denote a length-$n$ vector. We define $\mathsf{Circ} : \mathbb{R}^n \to \mathbb{R}^{n \times n}$ as,*

$$\mathsf{Circ}(a) := \begin{bmatrix} a_1 & a_n & a_{n-1} & \cdots & a_2 \\ a_2 & a_1 & a_n & \cdots & a_3 \\ a_3 & a_2 & a_1 & \cdots & a_4 \\ \vdots & \vdots & \vdots & \ddots & \vdots \\ a_n & a_{n-1} & a_{n-2} & \cdots & a_1 \end{bmatrix}.$$

**Fact 3.3** (Folklore). *Let $a \in \mathbb{R}^n$ denote a length-$n$ vector. Let $\mathsf{Circ}$ be defined in Definition 3.2. Let $F \in \mathbb{C}^{n \times n}$ denote the discrete Fourier transform matrix. Using the property of discrete Fourier transform, we have*

$$\mathsf{Circ}(a) = F^{-1} \mathrm{diag}(Fa) F.$$

*We can thus multiply $\mathsf{Circ}(a)$ with an input vector of length $n$ in $O(n \log n)$ time using the Fast Fourier transform algorithm.*

## 3.4 Rescaled Circulant Matrix

Our algorithm will critically involve manipulating a certain kind of structured matrix we call a *rescaled circulant matrix*. In this section we define these matrices and prove basic properties which we will use.

**Definition 3.4** (Rescaled Circulant Matrix). *We say a square matrix $M \in \mathbb{R}^{n \times n}$ is rescaled circulant if there are diagonal matrices $D_1, D_2 \in \mathbb{R}^{n \times n}$ and a circulant matrix $C \in \mathbb{R}^{n \times n}$ such that $M = D_1 C D_2$.*

**Fact 3.5.** *If $M \in \mathbb{R}^{n \times n}$ is a rescaled circulant matrix (see Definition 3.4), then given as input a vector $v \in \mathbb{R}$, one can compute the matrix-vector product $Mv$ in $O(n \log n)$ time.*

*Proof.* Suppose $M = D_1 C D_2$, we first compute $D_2 v$ straightforwardly in $O(n)$ time. Then we compute $C \cdot (D_2 v)$ in $O(n \log n)$ time. Finallyt, we compute $D_1 \cdot (C D_2 v)$ in $O(n)$ time. □

**Lemma 3.6.** *If $A$ and $B$ are rescaled circulant matrices, then $A \circ B$ is also a rescaled circulant matrix.*

*Proof.* Suppose $A = \text{diag}(a_1) A_2 \text{diag}(a_3)$ where $A_2$ is a circulant matrix.

Suppose $B = \text{diag}(b_1) B_2 \text{diag}(b_3)$ where $B_2$ is a circulant matrix.

We can show

$$
\begin{aligned}
A \circ B &= (\text{diag}(a_1) A_2 \text{diag}(a_3)) \circ (\text{diag}(b_1) B_2 \text{diag}(b_3)) \\
&= \text{diag}(a_1) \text{diag}(b_1)((A_2 \text{diag}(a_3)) \circ (B_2 \text{diag}(b_3))) \\
&= \text{diag}(a_1) \text{diag}(b_1)(A_2 \circ B_2) \text{diag}(a_3) \text{diag}(b_3).
\end{aligned}
$$

Therefore, we know $A \circ B$ is also a rescaled circulant matrix. $\square$

**Lemma 3.7.** *If $A_1, \cdots, A_t$ are rescaled circulant matrices, then for any vector $v$, we have $(A_1 \circ A_2 \circ \cdots \circ A_t)v$ can be computed in $O(tn \log n)$ time.*

*Proof.* The proof directly follows from applying Lemma 3.6 and Fact 3.5, $t$ times. $\square$

# 4 HOW TO COMPUTE THE LINEAR ATTENTION UNDER RoPE

Before getting to RoPE softmax attention, in this section we address the simpler problem of computing RoPE *linear* attention, which does not have entry-wise $\exp$.

**Definition 4.1** (Linear Attention). *Let $S \subseteq [d] \times [d]$ denote a support and $|S| = O(d)$. Given $W_{-(n-1)}, \cdots W_{-1}, W_0, W_1, \cdots W_{n-1} \in \mathbb{R}^{d \times d}$ and for all $i \in \{-(n-1), \cdots, -1, 0, 1, \cdots, n-1\}$. Given $Q \in \mathbb{R}^{n \times d}$ and $K \in \mathbb{R}^{n \times d}$, $V \in \mathbb{R}^{n \times d}$.*

*We define matrix $A \in \mathbb{R}^{n \times n}$ such as follows*

$$
A_{i,j} := (\underbrace{Q_{i,*}}_{1 \times d} \underbrace{W_{i-j}}_{d \times d} \underbrace{K_{j,*}^\top}_{d \times 1}), \forall i \in [n], j \in [n]
$$

*We define $D := \text{diag}(A \mathbf{1}_n)$. The attention computation is going to output an $n \times d$ matrix*

$$
\underbrace{D^{-1}}_{n \times n} \underbrace{A}_{n \times n} \underbrace{V}_{n \times d}
$$

For this linear version, we now show how to reduce it to $O(|S|)$ Fast Fourier transforms (FFTs), each of which can be performed in $O(n \log n)$ time. Intuitively, our algorithm is going to write $A \in \mathbb{R}^{n \times n}$ in the form $A = \sum_{(l_1, l_2) \in S} B_{l_1, l_2}$ where $B_{l_1, l_2} \in \mathbb{R}^{n \times n}$ is a rescaled circulant matrix.

Recall the support $S$:

**Definition 4.2.** *Given a collection of weight matrices $W_{-(n-1)}, \cdots W_{-1}, W_0, W_1, \cdots W_{n-1}$, we use $S$ to denote their support such that $\forall i \in \{-(n-1), \cdots, n-1\}$, $\text{supp}(W_i) = S$.*

**Definition 4.3** (one-sparse matrix). *For each pair $(\ell_1, \ell_2) \in S$, and $i, j \in [n]$, define the matrix $W_{i-j}^{\ell_1, \ell_2} \in \mathbb{R}^{d \times d}$ to be all 0s except that entry $(\ell_1, \ell_2)$ is equal to $(W_{i-j})_{\ell_1, \ell_2}$.*

**Claim 4.4.** *Let one sparse matrix $W_{i-j}^{\ell_1, \ell_2} \in \mathbb{R}^{d \times d}$ be defined as Definition 4.3. Then,*

$$
W_{i-j} = \sum_{(\ell_1, \ell_2) \in S} W_{i-j}^{\ell_1, \ell_2}
$$

*Proof.* We can show that

$$
\begin{aligned}
W_{i-j} &= \sum_{(\ell_1, \ell_2) \in S} \underbrace{e_{\ell_1}}_{d \times 1} \underbrace{(W_{i-j})_{\ell_1, \ell_2}}_{\text{scalar}} \underbrace{e_{\ell_2}^\top}_{1 \times d} \\
&= \sum_{(\ell_1, \ell_2) \in S} W_{i-j}^{\ell_1, \ell_2}
\end{aligned}
$$

where the second step follows from Definition 4.3. $\square$

**Definition 4.5.** *For each pair $(\ell_1, \ell_2) \in S$, we define matrix $A^{\ell_1,\ell_2} \in \mathbb{R}^{n \times n}$ as follows:*

$$A^{\ell_1,\ell_2}_{i,j} := \underbrace{Q_{i,*}}_{1 \times d} \underbrace{W^{\ell_1,\ell_2}_{f(i-j)}}_{d \times d} \underbrace{K^{\top}_{j,*}}_{d \times 1}, \forall i \in [n], j \in [n]$$

**Claim 4.6.** *Let $A^{\ell_1,\ell_2} \in \mathbb{R}^{n \times n}$ be defined as Definition 4.5. Then, we can show*

$$A = \sum_{(\ell_1,\ell_2) \in S} A^{\ell_1,\ell_2}.$$

*Proof.* For each $i \in [n], j \in [n]$, we compute each $(i,j)$-th entry of matrix $A \in \mathbb{R}^{n \times n}$ as

$$\begin{aligned}
A_{i,j} &= Q_{i,*} W_{i-j} K^{\top}_{j,*} \\
&= Q_{i,*} \sum_{(\ell_1,\ell_2) \in S} W^{\ell_1,\ell_2}_{i-j} K^{\top}_{j,*} \\
&= \sum_{(\ell_1,\ell_2) \in S} Q_{i,*} W^{\ell_1,\ell_2}_{i-j} K^{\top}_{j,*} \\
&= \sum_{(\ell_1,\ell_2) \in S} A^{\ell_1,\ell_2}_{i,j}
\end{aligned}$$

where the second step follows from Claim 4.4, the third step follows from rearranging the summation, and the last step follows from the definition of $A^{\ell_1,\ell_2}_{i,j}$.

Thus, we complete the proof. $\square$

**Definition 4.7.** *Let $S$ be defined as in Definition 4.2. For each $(\ell_1, \ell_2) \in S$, we define matrix $C^{\ell_1,\ell_2} \in \mathbb{R}^{n \times n}$ as $C^{\ell_1,\ell_2}_{i,j} := (W_{i-j})_{\ell_1,\ell_2}$.*

**Claim 4.8.** *Let $A^{\ell_1,\ell_2} \in \mathbb{R}^{n \times n}$ be defined as Definition 4.5. We can show $A^{\ell_1,\ell_2} = \mathrm{diag}(Q_{*,\ell_1}) C^{\ell_1,\ell_2} \mathrm{diag}(K_{*,\ell_2})$.*

*Proof.* We can rewrite $A^{\ell_1,\ell_2}_{i,j}$ as follows

$$A^{\ell_1,\ell_2}_{i,j} = Q_{i,*} W^{\ell_1,\ell_2}_{f(i-j)} K^{\top}_{j,*} = Q_{i,*} e_{\ell_1} (W_{f(i-j)})_{\ell_1,\ell_2} e^{\top}_{\ell_2} K^{\top}_{j,*} = Q_{i,\ell_1} (W_{f(i-j)})_{\ell_1,\ell_2} K_{j,\ell_2}$$

We define $C^{\ell_1,\ell_2}_{i,j} = (W_{f(i-j)})_{\ell_1,\ell_2}$, then the above equation becomes

$$A^{\ell_1,\ell_2}_{i,j} = Q_{i,\ell_1} C^{\ell_1,\ell_2}_{i,j} K_{j,\ell_2}$$

Thus we can have

$$A^{\ell_1,\ell_2} = \mathrm{diag}(Q_{*,\ell_1}) C^{\ell_1,\ell_2} \mathrm{diag}(K_{*,\ell_2})$$

Therefore, we complete the proof. $\square$

**Claim 4.9** (Running Time). *Let matrix $A^{\ell_1,\ell_2} \in \mathbb{R}^{n \times n}$ be defined as Definition 4.5. For any vector $x \in \mathbb{R}^n$, we can compute $A^{\ell_1,\ell_2} x$ in $O(n \log n)$ time using FFT.*

*Proof.* Using Claim 4.8, we can show that $A^{\ell_1,\ell_2}$ is rescaled circulant matrix.

Since $A^{\ell_1,\ell_2}$ is a rescaled circulant, thus, for any vector $v$, we can compute $A^{\ell_1,\ell_2} v$ in $O(n \log n)$ time. $\square$

## 5 HOW TO HANDLE THE EXP TERMS

We now give our full algorithm for general RoPE attention. In Section 5.1, we study matrices which are the entry-wise products of a number of rescaled circulant matrix, and how to use that decomposition to quickly multiply such matrices with a vector. In Section 5.2, we show how to decompose the RoPE attention matrix into summation of a number of such structured matrices using the polynomial method. In Section 5.3, we show how to put everything together to get our main result.

## 5.1 THE RUNNING TIME OF HAMADARD PRODUCT OF RESCALED CIRCULANT MATRIX MULTIPLYING A VECTOR

**Lemma 5.1.** *Let $m : [d] \times [d] \to \mathbb{N}$ be any function[1]. Define the matrix $A^{(m)} \in \mathbb{R}^{n \times n}$ by*

$$A_{i,j}^{(m)} := \prod_{(\ell_1, \ell_2) \in S} (A_{i,j}^{\ell_1, \ell_2})^{m(\ell_1, \ell_2)}, \forall i \in [n], j \in [n].$$

*Then $A^{(m)}$ is also of the form rescaled circulant matrix (see Definition 3.4). Furthermore, for any vector $v \in \mathbb{R}^n$, $A^{(m)}v$ can be computed in $O((\sum_{(\ell_1, \ell_2) \in S} m(\ell_1, \ell_2)) \cdot n \log n)$ time.[2]*

*Proof.* We define set $S$ to be

$$\{(\ell_{1,1}, \ell_{2,1}), (\ell_{1,2}, \ell_{2,2}), \cdots, (\ell_{1,|S|}, \ell_{2,|S|})\} \subset [d] \times [d]$$

We define $t_i \in \mathbb{N}$ for each $i \in [|S|]$ as follows

$$t_i := m(\ell_{1,i}, \ell_{2,i}).$$

From the definition of $A_{i,j}^{(m)} \in \mathbb{R}$, we know that $A^{(m)} \in \mathbb{R}^{n \times n}$ can be written as the entry-wise product of a collection of matrices (where each matrix is a rescaled circulant matrix), i.e.,

$$A^{(m)} = (A^{\ell_{1,1}, \ell_{2,1}})^{\circ t_1} \circ (A^{\ell_{1,2}, \ell_{2,2}})^{\circ t_2} \circ \cdots \circ (A^{\ell_{1,|S|}, \ell_{2,|S|}})^{\circ t_{|S|}}$$

Using Lemma 3.6, we know the entry-wise product between any two rescaled circulant matrix is still a rescaled circulant matrix. Thus, applying Lemma 3.6 to the above equations for $\sum_{i=1}^{|S|} t_i$ times, we can show that $A^{(m)}$ is still a rescaled circulant matrix.

Using Lemma 3.7, we know that for any vector $v$, $A^{(m)}v$ can be computed in $O((\sum_{i=1}^{|S|} t_i) \cdot n \log n)$ time. $\qquad\square$

## 5.2 EXPANDING POLYNOMIALS INTO SUMMATION OF SEVERAL RESCALED CIRCULANT MATRICES

**Lemma 5.2.** *Let $M^1, \ldots, M^k \in \mathbb{R}^{n \times n}$ be rescaled circulant matrices. Let $p : \mathbb{R} \to \mathbb{R}$ be a polynomial of degree $\widetilde{d}$. Let $m \in \mathcal{M}$ be the set of functions $m : [k] \to \mathbb{N}$ such that $\sum_{\ell=1}^k m(\ell) \leq \widetilde{d}$. Consider the matrix $M \in \mathbb{R}^{n \times n}$ defined by $M_{i,j} := p(\sum_{\ell=1}^k M_{i,j}^\ell)$. Then $M \in \mathbb{R}^{n \times n}$ can be written as the following sum of rescaled circulant matrices:*

$$M = \sum_{m \in \mathcal{M}} \alpha_m \cdot N^{(m)}$$

*Here $N^{(m)} \in \mathbb{R}^{n \times n}$ is defined as $N_{i,j}^{(m)} = (M_{i,j}^\ell)^{m(\ell)}$ for all $i \in [n], j \in [n]$ and $\alpha_m \in \mathbb{R}$ is coefficient. Furthermore, the number of rescaled circulant matrices is $|\mathcal{M}| = O(\binom{\widetilde{d}+k}{k})$.*

*Proof.* Recall $\mathcal{M}$ is the set of functions $m : [k] \to \mathbb{N}$ such that $\sum_{\ell=1}^k m(\ell) \leq \widetilde{d}$. Then, for each $m \in \mathcal{M}$ there is a coefficient $\alpha_m \in \mathbb{R}$ such that we can rewrite polynomial $p$ as follows:

$$p(z_1 + \cdots + z_k) = \sum_{m \in \mathcal{M}} \alpha_m \cdot \prod_{\ell=1}^k z_\ell^{m(\ell)}. \tag{2}$$

Thus,

$$M_{i,j} = p(\sum_{\ell=1}^k M_{i,j}^\ell) = \sum_{m \in \mathcal{M}} \alpha_m \cdot \prod_{\ell=1}^k (M_{i,j}^\ell)^{m(\ell)} = \sum_{m \in \mathcal{M}} \alpha_m \cdot N^{(m)}$$

where the first step follows from definition of $M$, the second step follows from Eq. (2), and the last step follows from definition of $N^{(m)}$. Thus, we can see $M = \sum_{m \in \mathcal{M}} \alpha_m \cdot M^{(m)}$.

$\qquad\square$

---

[1] Here intuitively, $m$ represents the exponents of variables in a monomial of a polynomial.

[2] Later, we will show that $\sum_{(\ell_1, \ell_2) \in S} m(\ell_1, \ell_2) = n^{o(1)}$ for the function $m$ we used in this paper.

## 5.3 MAIN RESULT

Finally, we are ready to put all our techniques together.

**Theorem 5.3** (Restatement of Theorem 1.3). *Suppose $d = O(\log n)$ and $B = o(\sqrt{\log n})$. There is an $n^{1+o(1)}$ time algorithm to approximate* ARAttC *up to $\epsilon = 1/\operatorname{poly}(n)$ additive error.*

*Proof.* We use the polynomial of Lemma 3.1 in Lemma 5.2 with choice of $k = |S| = O(d) = O(\log n)$ and $\widetilde{d} = o(\log n)$ is the degree of the polynomial from Lemma 3.1 for error $1/\operatorname{poly}(n)$. We can thus upper bound

$$|\mathcal{M}| = O(\binom{k+d}{d})) = n^{o(1)}.$$

The total running time consists of three parts: first, approximating $A\mathbf{1}_n$ which gives an approximation to diagonal matrix $D$; second, approximating $Av$ for $d$ different columns vectors $v$, this will approximate $AV$; third, combining approximation of $D^{-1}$ with approximation of $AV$, to obtain an approximation of $D^{-1}AV$. Combining Lemma 5.1 and 5.2. The dominating running time for above three parts is

$$|\mathcal{M}| \cdot \sum_{(\ell_1, \ell_2) \in S} m(\ell_1, \ell_2) \cdot n \log n = O(n^{1+o(1)})$$

Due to the choice of $|\mathcal{M}| = n^{o(1)}$, $|S| = O(d)$, $d = O(\log n)$.

The error analysis remains identical to prior attention algorithms using the polynomial method (Alman & Song, 2023), thus we omit the details here. □

## 6 HARDNESS

Before we state our lower bound, we present The Strong Exponential Time Hypothesis. The Strong Exponential Time Hypothesis (SETH) was introduced by Impagliazzo and Paturi Impagliazzo & Paturi (2001) over 20 years ago. It is a strengthening of the P $\neq$ NP conjecture, which asserts that our current best SAT algorithms are roughly optimal:

**Hypothesis 6.1** (Strong Exponential Time Hypothesis (SETH)). *For every $\epsilon > 0$ there is a positive integer $k \geq 3$ such that $k$-SAT on formulas with $n$ variables cannot be solved in $O(2^{(1-\epsilon)n})$ time, even by a randomized algorithm.*

SETH is a popular conjecture which has been used to prove fine-grained lower bounds for a wide variety algorithmic problems, as discussed in depth in the survey Williams (2018).

**Theorem 6.2** (Restatement of Theorem 1.4). *Assuming SETH, for every $q > 0$, there are constants $C, C_a, C_b > 0$ such that: there is no $O(n^{2-q})$ time algorithm for the problem* ARAttC$(n, d = C\log n, B = C_b\sqrt{\log n}, \epsilon = n^{-C_a})$.

*Proof.* We will pick all of the $W_{-(n-1)}, \cdots, W_{(n-1)} \in \mathbb{R}^{d \times d}$ to be an identity $I_d$ matrix. Thus the RoPE attention becomes classical attention. Thus using Alman & Song (2023), our lower bound result follows. □

## 7 CONCLUSION

In this work, we provide an almost linear time algorithm for RoPE attention. RoPE attention is used as a more expressive variant on attention in many applications, but the usual polynomial method approach inherently cannot work for calculating it quickly. We introduced a new way to combine the polynomial method with our "rescaled circulant matrices" and the Fast Fourier transform in order to solve this problem more efficiently. As future work introduces more variants on attention, it will be exciting to explore whether these and other linear algebraic tools can still be used to perform fast computations.

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
