# OpenReview forum: "Fast RoPE Attention: Combining the Polynomial Method and Fast Fourier Transform"
_ICLR.cc/2025/Conference — Submitted to ICLR 2025_

### Official Review · Reviewer_sE6r · 2024-10-18

**Soundness:** 1
**Presentation:** 4
**Contribution:** 3
**Rating:** 3
**Confidence:** 4

**Summary:**

The paper presents an algorithm, along with its complexity analysis, to efficiently approximate a variant of attention called RoPE attention, which is used in recent LLMs. The algorithm is based on approximating softmax with a rational function, computing efficiently polynomials of circulant matrices, and showing that attention can be recasted in this framework. The complexity of the algorithm matches the lower-bound for this problem.

**Strengths:**

The paper is very clearly-written and the algorithm seems correct. The algorithm is novel and combines nicely several ideas from applied mathematics (approximation of exp by a polynomial, FFT to compute the product of a circulant matrix and a vector). It tackles the crucial question of efficiently computing attention matrices, and the algorithm could therefore have good impact. I stand in favour of acceptance, and encourage authors to complement their contribution with pseudo-code, open-source implementation, and experiments (see Weaknesses).

**Weaknesses:**

[POST-REBUTTAL EDIT: Reviewer Q6JF found a very significant issue in the paper. Namely, some matrix central to the analysis is supposed to be circulant, whereas in the case of standard RoPE attention, this matrix is only constant on each diagonal, which is weaker than circulant. Thus, the proposed algorithm probably does not apply to standard RoPE attention. The authors did not provide an answer during the rebuttal (see discussion below the review of Reviewer Q6JF). Thus I cannot recommend acceptance of the paper, and lowered my score.]

- Section 5.3 is a bit hard to follow and could be more detailed. For instance, a self-contained algorithm in pseudo-code would be beneficial both for the overall understanding of the paper, and for checking correctness of the proof of Section 5.3.
- There is no experiment. While I don’t think that experiments are an absolute prerequisite for a ML paper, it would be nice to provide an open-source implementation of the algorithm and to compare empirically the performance in terms of speed and accuracy with a naive computation of RoPE attention (and perhaps other baselines if existing), and with the computation of standard attention with the polynomial method, for various hyperparameter values. Given the nature of the contribution, I am still in favor of acceptance despite the absence of experiments, but encourage authors to provide experiments in the next version of the paper.

Minor remarks and typos (no influence on the rating):
- line 67: I believe $B$ is not defined at this point.
- line 304: please find a proper reference instead of “Folklore”.
- line 380 up to the end of Section 4: there are $f$ several times in indices of $W$. Is this a typo?
- in Lemma 5.1, specify again that the $A^{\ell_1, \ell_2}$ are rescaled circulant matrices.
- In Lemma 5.2, rephrase the definition of $\mathcal{M}$: “let $\mathcal{M}$ be the set of functions …”
- In the proof of Lemma 5.2, recall again why $N^{(m)}$ are rescaled circulant matrices (due to Lemma 3.6).
- In the proof of Lemma 5.2, explain where the estimate of $|\mathcal{M}|$ comes from.
- line 482: $M^{(m)}$ should be $N^{(m)}$.
- line 497: $d$ should be $\tilde{d}$?

**Questions:**

- To the best of your knowledge, is the polynomial method actually used in practice to approximate the attention computation in large-scale Transformer models?
- The number of circulant matrices increases from $|S|$ to $|\mathcal{M}|$ due to the element-wise product by the polynomial $p$. This means that, if we apply the algorithm several times (across several layers of Transformer), the number of matrices to handle grows at each iteration. Thus the complexity of applying recursively, say $L$ times, this algorithm, could grow super-linearly with respect to $L$. Do you know this complexity?

---

> ### Author Response · Authors · 2024-11-22
>
> Thank you for the thorough review. We will implement the suggested changes in the final version.
>
> To answer your questions:
> - A number of papers have been published about empirically evaluating the polynomial method for attention, including by researchers at companies which have very large-scale Transformer models. (One example is PolySketchFormer.)
> - The final running time given here is performing a single RoPE attention computation, so if one were performing $L$ different computations (either in parallel, or in series, or otherwise over the course of a larger Transformer computation), the running time should scale linearly with $L$, since it would just perform this algorithm $L$ times. (Sorry in advance if we're misunderstanding the question.)

---

> > ### Comment · Reviewer_sE6r · 2024-11-22
> > **Thank you**
> >
> > Thank you for your answer. Regarding the depth, you indeed missed the point of my question. Obviously, doing $L$ times the same computation multiplies the complexity by $L$. However, in the case of deep Transformer, the computations are applied recursively, and the input size increases from one step to the next (due to the increase in the number of circulant matrices to handle at each step, cf my initial question). This makes it less obvious. For instance, if the complexity at a given step is $O(n)$ where $n$ is the input size, and the input size doubles at each step, the total complexity is $\sum_{\ell=0}^{L-1} 2^\ell n = \Theta(2^L n)$.

---

### Official Review · Reviewer_FQfH · 2024-10-28

**Soundness:** 3
**Presentation:** 2
**Contribution:** 3
**Rating:** 5
**Confidence:** 2

**Summary:**

The paper discusses an approach to approximating RoPE attention, building on previous work that approximated standard attention using the polynomial expansion of exp. The work deals with Rope using FFT and shows how to extend previous results relatively easily, including results on approximation performance.

**Strengths:**

Approximating attention is an important topic and these results go some way to state how well RoPE attention can be approximated quickly. As far as I know, these results are new and do add to knowledge about how well we can approximate Rope attention.

**Weaknesses:**

The paper isn't particularly clear in places, with some definitions out of sequence. The results also seem quite straightforward though there are also places where more justification or explanation seems required. The lack of any empirical performance is a shame since I think most practitioners would like to see whether the theory actually translates into a practically useful approximation.

**Questions:**

*** page 1

The attention should be normalized by \sqrt(d), not d.

*** page 2

B is not defined

*** page 3

equation 1 has no normalization, yet definition 1.1 does have normalization (though this should be sqrt(d), not d)

The notation [n] is perhaps well used in combinatorics, but is not common elsewhere. Please help the reader and explain this notation.

After reading through, the authors define this on page 6, but this is too late for the reader.

*** page 4

The argument that A cannot be low rank isn't very clear to me and I feel the explanation could be improved. For example, one can choose a rotation R that is the identity, recovering the standard attention (which can have low rank).

I find the presentation based on discussing whether M "were indeed circulant a matrix" confusing. Clearly, M cannot in general be circulant (since the exponent in equation 1 is not circulant) and this discussion I feel confuses rather than helps the reader.

*** page 6

My immediate thought is that the exponential can only be well approximated for small x. Lemma 3.1 seems to confirm that the degree of the polynomial (and hence complexity of the approximation) scales with the range of x. The immediate question is then whether this would be practical in real-case scenarios.

Fact 3.3. This is called "folklore". Please cite a reference to this result. "Folklore" suggests that it may not be true, whilst this is indeed a fact.

By definition a circulant matrix multiplied by a vector \sum_j A_{i-j}v_j is a convolution and therefore can be calculated using the FFT in O(n\log n).

line 323 -- typos

*** page 7

The notation in line 351 is confusing. This W isn't the same W as in the definition of the original attention

Definition 4.1 introduces S but doesn't use it in the definition.

In general I found the presentation in section 4 quite confusing. If one wants to compute \sum_j A_{i,j}v_j where we define the linear attention A_{i,j} = vec(q_i)\trans R^{j-i} vec(k_j) for the ith query vector vec(q_i) and jth key vector vec(k_j) and rotation matrix R raised to the power j-i, then trivially we can write \sum_j A_{i,j}v_j= vec(q_i)\trans\sum_j R^{j-i}vec(k_j)v_j. Since \sum_j R^{j-i}vec(k_j)v_j is a convolution, then it directly follows that Av is computable in O(n\log n) time. It would be useful if the authors could therefore explain why the circulant results are required.

*** page 10

The justification for theorem 5.3 isn't fully clear to me. The previous results relate to approximating the unnormalised attention. However, we need to divide the approximated unnormalised attention by the approximated normalised attention. I don't immediately see why the division doesn't affect the approximation error results. This could at least be better explained.

The hardness result seems essentially trivial.

*** General comments

It's disappointing that there is no empirical support for the method including, for example, a simple experiment to show how well the method works.

---

> ### Author Response · Authors · 2024-11-22
>
> Thank you for the very thorough review. We will implement all the suggested changes, and wish to comment on just a few:
>
>
> **normalizations** You're right that the normalizations are important, and our results are indeed for the correct normalization by \sqrt{d} (we will correct this typo). We ultimately use the same normalization/error analysis as Alman & Song (2023), which uses the proper normalization.
>
> **Section 4** You're right that the approach you give would also work for linear attention. However, this asymmetric approach (i.e., which treats i and j differently) ends up generalizing poorly in section 5 once one incorporates the entry-wise exponentiation. In short, when one gets down to line 497, it would replace d with d^2 in the binomial coefficient, and thus result in a worse rank bound which only applies for smaller B.
>
> **empirical results** We agree that an empirical evaluation of our algorithm would be exciting future work. That said, we believe the theoretical results we provide here are more interesting than an empirical evaluation for two reasons.
>
> First, since RoPE attention appears more complicated than regular attention, we previously believed it would be possible to prove a greater lower bound for RoPE attention. Our algorithm shows this intuition is wrong, and an improved lower bound is not possible, since the same approximation can be achieved quickly for RoPE attention. This main message, that RoPE attention is no harder than attention in theory, was very surprising to us.
>
> Second, many different works have explored different ways to make use of positional information in attention and attention variants. One could empirically evaluate a particular version of RoPE attention, and a new variant would likely be introduced the very next day. However, our new principled way to combine the polynomial method and FFT should work for any such variant, including new ones that researchers develop in the future, and can hopefully guide future approaches to incorporating positional information.

---

> > ### Comment · Reviewer_FQfH · 2024-11-24
> >
> > Thanks for the response. My sense is that this is potentially a very interesting piece of work. However, without an updated version to examine, it's difficult to be confident that the paper will be in a form that would be more palatable, particularly since I feel that the presentation is letting the paper down at the moment. In that sense, I think the paper would benefit from a revision and doesn't quite pass the threshold for acceptance.

---

### Official Review · Reviewer_pwfx · 2024-10-28

**Soundness:** 3
**Presentation:** 2
**Contribution:** 1
**Rating:** 5
**Confidence:** 4

**Summary:**

The paper presents a sublinear algorithm for efficient Rotary Position Embedding (RoPE) attention computation with provable guarantees. The main technique combines the polynomial method with the Fast Fourier Transform (FFT). The paper also presents a lower bound on this problem.

**Strengths:**

- Rotary positional encoding is very popular in practice. It is relevant and timely to consider efficient algorithms for RoPE-attention.

- The paper presents the main technique clearly.

**Weaknesses:**

- **Limited algorithmic novelty.** The proposed method relies on the polynomial method to handle $\exp$ in attention, which has been studied in existing work [1,2]. Fast Fourier Transform has also been introduced in prior works to efficiently handle positional information in attention [3,4,5]. While this paper studies attention with a different positional encoding variant (RoPE), it would enhance the paper quality if the authors can discuss more on the new algorithmic insights in the proposed method.

[1] On The Computational Complexity of Self-Attention

[2] Fast attention requires bounded entries

[3] Stable, Fast and Accurate: Kernelized Attention with Relative Positional Encoding

[4] Toeplitz Neural Network for Sequence Modeling

[5] From block-Toeplitz matrices to differential equations on graphs: towards a general theory for scalable masked Transformers

- **Weak lower bound.** The lower bound only works for the "_general_" RoPE Attention Computation. The hard instance is constructed by setting all the $W_i$'s to identity matrices, falling back to the vanilla attention. This results does not lead to any new insight on the hardness of approximating the true RoPE attention in practice. It is not clear whether the exact RoPE attention (with $R_{j-i}$ defined as in line 111) satisfies similar hardness results.

- **Lack of empirical evaluations.** The authors do not provide any implementation of the proposed algorithm and run experiments on any datasets. It may help the practitioners if the implementation and empirical analysis on the speed-up rate, modeling quality, and comparisons with other efficient attention methods can be provided. Empirical evaluations can also better justify the theoretical results.

- **Minor issues.**
  - `\citep` and `\citet` are not used correctly in the paper, e.g., in lines 58, 88, 89, 90.
  - Line 116: The word "about" is unnecessary and makes the statement vague. In RoPE, $\alpha=10^4$ exactly.
  - Def. 1.1 is about "A General Approximate RoPE Attention Computation", but in its statement the terminology changes to "Rotated attention computation". Please make it consistent.
  - Def 4.2 should have been stated before Def. 4.1 to make the exposition logical. Also, should $\mathrm{supp}(W_i)=S$ be $\mathrm{supp}(W_i)\subset S$ instead?

**Questions:**

- In Def. 1.1, should the scaling factor be $1/\sqrt{d}$ instead of $1/d$?

- Can you discuss whether the assumptions on $B$ (the upper bound on queries, keys, and values) hold in practice based on empirical evidence?

- Do you have evidence to show whether the assumption $B=o(\sqrt{\log n})$ holds in practice? Or, under what condition is this assumption more likely to be violated?

---

> ### Author Response · Authors · 2024-11-22
>
> Thank you for your thorough review.
>
> **algorithmic novelty**: You're right that the polynomial method and FFT have both been used previously for attention. We also discuss this on page 5. That said, we believe we are the first to **combine** these two approaches in a single algorithm. See the section on page 4 called "Technique Overview: Limitation of Prior Techniques" where we discuss the barriers to using prior approaches for RoPE attention, and the subsequent section on page 4 where we describe the novel way we combine these techniques.
>
> **lower bound** You're right that we're proving a lower bound on a special case of the general RoPE attention here (albeit, the case which appears the easiest). In fact, one can modify the lower bound proof for attention of Alman & Song, 2023 to work for the usual RoPE attention as well. The modification is straightforward but tedious, so we previously didn't think it was worth including. That said, you are absolutely correct that it would strengthen the lower bound statement, so we will include it in the next version.
>
> **implementations** We agree that an empirical evaluation of our algorithm would be exciting future work. That said, we believe the theoretical results we provide here are more interesting than an empirical evaluation for two reasons.
>
> First, since RoPE attention appears more complicated than regular attention, we previously believed it would be possible to prove a greater lower bound for RoPE attention. Our algorithm shows this intuition is wrong, and an improved lower bound is not possible, since the same approximation can be achieved quickly for RoPE attention. This main message, that RoPE attention is no harder than attention in theory, was very surprising to us.
>
> Second, many different works have explored different ways to make use of positional information in attention and attention variants. One could empirically evaluate a particular version of RoPE attention, and a new variant would likely be introduced the very next day. However, our new principled way to combine the polynomial method and FFT should work for any such variant, including new ones that researchers develop in the future, and can hopefully guide future approaches to incorporating positional information.
>
> **questions**
> - Yes you're right, thanks for noting the typo.
> - See the paragraph called "Bounded entries in practice" on page 2. In short, a number of prior works have used techniques to bound the entries or observed that entries are already bounded in order to speed up algorithms.
> - The papers discussed in the paragraph on "Bounded entries in practice" on page 2 give algorithmic techniques for imposing that the bound on $B$ must hold. In those papers, they find that imposing this bound only slightly hurts the quality of the outputs of the language models.

---

> > ### Comment · Reviewer_pwfx · 2024-11-24
> > **Response to author rebuttals**
> >
> > I thank the authors for the rebuttal. After reading the authors' rebuttal, several concerns remain:
> > - Based on my understanding of the discussion in page 4, the technique for incorporating RoPE is to leverage the property of rescaled circulant matrices (which supports efficient computation via FFT). This idea has been well-established for other variants of positional encoding as mentioned in my original review. The only novelty here, if any, is to adapt it to RoPE.
> > - While the authors claim that proving the lower bound for the original RoPE attention is straightforward, I don't see any updated proof in the submission. Thus, this concern is not addressed.
> > - The authors' justification on no empirical evaluation is not satisfactory either.
> >   - First, I do not share the same surprise with the authors on the theoretical results, given many existing works on handling structured positional encoding matrices via FFT in $n\log n$ time.
> >   - Second, since the author claims their techniques are highly general, it is natural to expect an implementation with an interface that could support any kind of RoPE attention in the framework. While the authors can evaluate the method on one particular variant, the implementation can still facilitate future researchers to explore other RoPE variants as the authors hoped.
> >   - Moreover, I do not think the authors have carefully investigated the practical applications of RoPE. Today's popular LLMs, including Llama and Gemma, use RoPE as was originally proposed with no fancy modifications at all. Thus, the possibility of new variants in the future is not an excuse for no experiments, and evaluation on the original RoPE is meaningful enough.
> >
> > Therefore, I decide the maintain my ratings and recommend a rejection.

---

### Official Review · Reviewer_Q6JF · 2024-11-04

**Soundness:** 2
**Presentation:** 2
**Contribution:** 2
**Rating:** 3
**Confidence:** 4

**Summary:**

The paper, essentially, establishes the following result: insertion of a specific (i,j) position-dependent square dxd matrix **R** between queries matrix **Q** and keys matrix **K** in the attention mechanism **A**  such that  $A_{ij}=exp(Q_i R_{i-j} K_j^\top)$ leads to an algorithm which under assumption of bounded norm in attention parameter matrices **Q, K, V** computes the approximate attention in $O(n \log n)$ time. They call $R_{i-j}$ *general RoPE* matrix and consider specific constructions which makes the resulting **A** matrix a *rescaled circulant matrix* which could be constructed and multiplied by any other vector in $O(n \log n)$ by means of Fast Fourier Transform.

**Strengths:**

* The paper reveals interesting and promising direction of work on optimization of runtime of Transformers by injecting specific types of structured matrices.
* Their approach is inspiringly novel and non-trivial and, if studied further, could lead to novel augmented Transformer architectures having the same quality as the original but computable in $O(n \log n)$ time.
* For general setting when the proposed generalized RoPE matrix can't be reformulated as entry-wise transformation of **Q** and **K**, the method is sound as it seems to be capable of computing approximate attention in sub-quadratic run-time complexity with guaranteed level of precision under some assumptions.

**Weaknesses:**

Unfortunately, several key claims and assumptions in the paper are mistaken.

1. Theorems 1.3 and 5.3 are not correct. The statements clearly articulate the $O(n^{1+o(1)})$ runtime of the proposed algorithm while the proofs directly show that the algorithm requires $O(n \log n)$ operations which is greater than $O(n^{1+o(1)})$.
2. It invalidates one of  the main claims of the paper (lines 155-159) that their algorithm is in the same complexity class as the less complicated previous algorithms [1, 2] for approximate softmax attention computation.
3. Theorems 1.5 and 5.2 are not complete. The proof considers only the case when $R_{i-j}=I$ (identity matrix). What about more general case, can it be reduced to the identity matrix?
4. The claims (lines 182-188) that in standard RoPE attention "the underlying matrix which exp is applied to no longer needs to have low rank" and the prior fast approximation techniques "fundamentally cannot apply to RoPE attention"  seem to be poorly worded and mistaken. Attention matrix $A \in \mathbb{R}^{n \times n}$ is a product of  **Q** and **K** of size (n, d) each, thus having rank at most d (d<<n). Applying the matrix projection $R \in \mathbb{R}^{d \times d}$ to either $Q$ or $K^\top$ does not change their dimensions, so **A** still maintains its rank. The same is true when a vector  $Q_i$ or $K_j$ is projected onto $R_{i-j}$, and then all $n$ vectors are stacked to form the entire **Q** or **K** matrix.
5. Moreover, usually RoPE are applied efficiently in entry-wise fashion with $O(nd)$ time as opposed to $O(nd^2)$ time when $R_{i-j}$ is a (d, d)-size matrix. In conjunction with previous point, it weakens the cause for implementing the algorithm, at least for standard RoPEs.
6. The authors state that standard RoPE are a special case of their proposed general RoPE matrices. They also state that any sequence of $R_{i-j}$ realizations can be constructed in such a way that the resulting pre-softmax attention becomes a rescaled circulant matrix (lines 186-188). They expand on it in Claim 4.8-4.9 as equation in line 415 can characterize  a rescaled circulant matrix if $C^{l_1, l_2}$ is a (possibly rescaled) circulant matrix.

However, it's not possible to choose underlying elements $l_1, l_2$ of $R_{i-j}$ in case of standard RoPE in such a way that $C^{l_1, l_2}$ is a (rescaled) circulant matrix. To see that, construct matrix $C^{1, 1}$ with $C_{i,j}^{1, 1}=\cos (\theta (i-j))$. Its elements depend only on difference of positions (i-j). It's a symmetric matrix with an additional structure which differs from circulant matrix structure. Neither its possible to express this matrix as a rescaled circulant matrix for any sequence length n in general.

Thus, standard RoPE cannot be viewed as circulant-matrix inducing structure, and we get a contradiction.

7. In line 187 authors make an even stronger claim that "by picking the Rj−i entries appropriately, one can choose M to be any circulant matrix" (not merely *some rescaled* circulant matrix  but *any fully* circulant matrix) which is not further proved or discussed.

References:
[1] Josh Alman and Zhao Song. Fast attention requires bounded entries. In NeurIPS, 2023.

[2] Feyza Duman Keles, Pruthuvi Mahesakya Wijewardena, and Chinmay Hegde. On the computational complexity of self-attention. In International Conference on Algorithmic Learning Theory, pp. 597–619. PMLR, 2023.

**Questions:**

See weaknesses.

---

> ### Author Response · Authors · 2024-11-22
>
> Thank you for the review. We strongly believe that all the claims and assumptions in our paper are correct.
>
> 1. The Theorem statements are correct as written. The relationship between $n \log n$ and $n^{1 + o(1)}$ is the opposite of what you claim. Since, for every $\varepsilon>0$, we have $n \log n < n^{1 + \varepsilon}$ for big enough $n$, it follows that $n \log n \leq n^{1 + o(1)}$. See, for instance, https://cs.stackexchange.com/a/152315
> 2. We're not sure what claim you're saying is invalidated here. This is actually the main point of our paper: that although RoPE attention appears more complicated than regular attention, we are actually able to give a new algorithm to approximately compute it in the same running time.
> 3. We believe you're referring to Theorems 1.4 and 6.2, rather than Theorems 1.5 and 5.2? At any rate, these theorems are also correct as written. We observe that the ARAttC problem is hard even in the special case when all the W matrices are the identity matrix, and thus it is not possible to design a faster algorithm for ARAttC.
> 4. Our claim is correct. Entry-wise exponentiating a matrix does not maintain its rank as you suggest, and this is why the usual attention cannot be computed exactly in almost linear time. As a simple example, the following $2 \times 2$ matrix has rank 1, but when you entry-wise exponentiate it, it has rank 2: $\begin{bmatrix}
> 1 & 2 \\\\
> 2 & 4
> \end{bmatrix}$
> https://www.wolframalpha.com/input?i=rank+of+%7B%7Be%5E1%2C+e%5E2%7D%2C+%7Be%5E2%2C+e%5E4%7D%7D
> 5. We're not sure what algorithm you're referring to here, but the straightforward algorithm takes $\Theta(n^2 d)$ time since it constructs the $n \times n$ attention matrix, which cannot be done in subquadratic time in $n$.
> 6. You observe that $C^{1,1}$ has the property that "Its elements depend only on difference of positions (i-j)". This is exactly the definition of a circulant matrix! Thus, your subsequent claim that "it's a symmetric matrix with an additional structure which differs from circulant matrix structure" is incorrect, and this is not in any way a counterexample.
> 7. When $d=1$, each $R_{j-i}$ is a scalar value, and $M$ is the matrix whose $(i,j)$ entry is the value $R_{j-i}$. You can thus pick the scalars so that $M$ is any circulant matrix you would like.

---

> ### Comment · Reviewer_Q6JF · 2024-11-22
>
> 1) Unfortunately, I cannot agree that computational complexity $n \log n \leq n^{1+o(1)}$. The formal definition of o(1) function g(n) is $ \lim_{n \to \infty} \frac{g(n)}{1} = \lim_{n \to \infty} g(n) = 0$. The same is stated in the answers of the StackExchange questions you provided (https://cs.stackexchange.com/a/152315). Please also note that, while $n \log n \leq n^{1+\epsilon}$ is a true statement, any constant positive function $g(n)=\epsilon$ is not in o(1) class. So, $ \lim_{n \to \infty} n^{1+o(1)} = \lim_{n \to \infty} n^{1+0} = \lim_{n \to \infty} n \le  \lim_{n \to \infty} n \log n$.
>
> 2) The issue from point 1 invalidates of the main claims of the paper (lines 155-159) that the algorithm is in the same runtime complexity class ($n^{1+o(1)}$ to be specific) as in paper [1], which you build upon, and [2]. The paper [1] establishes a truly $n^{1+o(1)}$ algorithm. If you compare your algorithm with plain exact attention calculation , which has $O(n^2)$ runtime complexity, and not with the approximate algorithm from [1], please state it explicitly. As of now, the phrasing "we are able to achieve the same running time for this more expressive version" in conjunction with claimed $n^{1+o(1)}$ complexity indicates comparison with [1].
>
> 3) Thank you for the correction, I referred to Theorems 1.4 and 6.2, indeed. The special case when all the W matrices are identity matrix is the case of the regular attention, which is proved in [1]. But the general case of RoPE attention in your work when the W matrices are *not* identity matrix is not addressed. In order to prove the Theorem, you should show that all other cases can be reduced to this special case.
>
> 4) In the paper you talk about pre-exponentiated attention matrix (M in your notation from lines 182-188), to quote: "the underlying matrix which exp is applied to no longer needs to have low rank" and "since M itself is not low-rank."
>
> 5) I refer to the standard RoPE calculation algorithm which runs in O(nd) time (see, for example, https://github.com/huggingface/transformers/blob/main/src/transformers/models/llama/modeling_llama.py#L203). You suggest using $O(nd^2)$ algorithm for the same operation, which is impractical, at least in case of standard RoPEs.
>
> 6) Consider a matrix $C^{1, 1}$ in case of standard RoPE and sequence length n=3:
> $
> \begin{bmatrix}
> a & b & c \\\\
> b & a & b \\\\
> c & b & a
> \end{bmatrix}
> $, where $a = \cos 0\theta$,  $b = \cos 1\theta$, and $c = \cos 2\theta$. This is clearly not a circulant matrix.
>
> 7) Unfortunately, this claim does not make sense for me.

---

> > ### Comment · Reviewer_sE6r · 2024-11-23
> > **Incorrect statements by reviewer Q6JF**
> >
> > I would like to clarify that several statements of reviewer Q6JF are incorrect. It is important we ensure such statements are accurate to maintain a fair and constructive review process.
> >
> > - 1. and 2. It is true that $n \log(n) = n^{1 + o(1)}$. Indeed $\log(n) = n^{u_n}$ iff $u_n = \log(\log n) / \log(n)$, which does tend to zero.
> >
> > - 3. To prove a complexity lower-bound over a class of problems, it is sufficient to exhibit one problem instance where no algorithm succeeds.
> >
> > - 4. The matrix $M$ can be full rank. It is not defined as a product of low-rank matrices. For example, in the case where $d=1$, $R_0 = 1$, $R_k = 0$ for $k \neq 0$, $X = (1, ..., 1)$, $W_k = W_q = 1$, we get $M=I$.
> >
> > - 6. The matrix given the reviewer is circulant. More generally, the definition of $C^{1,1}$ implies it is circulant.

---

> > > ### Author Response · Authors · 2024-11-26
> > >
> > > Thank you Reviewer Q6JF for your responses, and also Reviewer sE6r for weighing in.
> > >
> > > We'd like to respond to Reviewer Q6JF's most recent comment and hopefully clear up any remaining confusion.
> > >
> > > 1. In your comment, the step with an issue is where you write $\lim_{n \to \infty} n^{1+o(1)} = \lim_{n \to \infty} n^{1+0}$. Since $n \to \infty$, you cannot apply the limit just to the exponent $o(1)$ without taking into account the base $n$ which is also growing.
> > > In our case, as Reviewer sE6r pointed out, we consider the function $f(n) = \frac{\log(\log n)}{\log n}$. We can see that $\lim_{n \to \infty} f(n) = 0$, meaning $f(n) < o(1)$, and so $n^{1 + f(n)} < n^{1 + o(1)}$. Meanwhile, $n^{f(n)} = 2^{\log (n) \cdot f(n)} = 2^{\log(\log n)} = \log n$, and so $n^{1 + f(n)} = n \log n$. We thus have as desired that $n \log n < n^{1 + o(1)}$.
> > >
> > > 2. Indeed, the point of the sentence at line 155 is to say that, although RoPE attention appears more complicated than the usual attention, we are able to achieve the same $n^{1 + o(1)}$ running time guarantee in this paper as [1] is able to achieve for the usual attention. Earlier in the introduction is also building up this comparison; see lines 66-73 and lines 83-84.
> > >
> > > 3. As mentioned by Reviewer sE6r, in order to prove hardness for an algorithmic problem, it is not necessary to show that all possible instances are hard. It suffices to prove that some instances are hard. (Almost all hard problems have easy instances. For instance, even though we believe SAT is very hard in general, there are many easy SAT instances (such as CNF formulas where no literals are negated). One could thus not hope to show that all SAT instances are hard.) In this case, we prove hardness when W is the identity matrix, which correctly and completely proves our theorem statement. (Perhaps interestingly, this appears at first glance like it should be an easier instance of the problem but it is the instance we prove hardness for.)
> > >
> > > 4. Indeed, a key point is that $M$ can be a full rank matrix (even before the entry-wise exponentiation), as in the example of Reviewer sE6r. We also discuss this in more detail in the section on page 4 called "Technique Overview: Limitation of Prior Techniques".
> > >
> > > 5. We're still unsure about which part of our algorithm you're referring to here. Could you let us know specifically which line or page or theorem statement in our paper involves the $O(nd^2)$ running time that you mentioned? (In case it may ease your concern here, we'll also note that $nd^2 < n^{1+o(1)}$ since $d = O(\log n)$, and so even a running time of $O(nd^2)$ is negligible and would not impact the final running time of our algorithm.)
> > >
> > > 6. As observed by Reviewer sE6r, the matrix you gave in that example is indeed a circulant matrix. An easy way to spot that matrix is circulant is to check that all the entries along any diagonal are the same. In this case, for instance, all the entries on the main diagonal are equal to $a$, and all the entries on the diaogonal above that are equal to $b$.
> > >
> > > 7. When $d=1$, each $R_{j-i}$ is a $1 \times 1$ matrix, i.e., it is a single number. The matrix $M$  is the $n \times n$ matrix whose $(i,j)$ entry is equal to that number $R_{j-i}$. Now, consider, any $n \times n$ circulant matrix $C$ that you would like. Let's show that we can pick the $R_{j-i}$ values so that $M=C$. Since $C$ is circulant, by definition, there is a function $t : \mathbb{Z} \to \mathbb{R}$ such that, for all $i$ and $j$, the entry $(i,j)$ of $C$ is equal to $t(j-i)$. We can therefore pick $R_{j-i} = t(j-i)$. Then $M=C$ as desired.
> > >
> > > Please let us know if you have any further questions or confusions, and particularly if you could clarify specifically where in our paper your comment 5 is referring to. Thank you!

---

> ### Comment · Reviewer_Q6JF · 2024-11-26
> **Response to reviewer sE6r and the authors**
>
> I thank reviewer sE6r for the clarifications and would like to respond in details both to you and the authors.
>
> *1.* I agree that $n \log(n) = n^{1 + o(1)}$. Thank you for the proof. However, I believe notation $n^{1 + o(1)}$ is confusing and unintuitive and can lead to misinterpretations. For example, take two functions: $f_1(n) = \log(\log n) / \log(n)$ and $f_1(n) = \frac{1}{n}$ which tend to zero and, thus, $o(1)$. Then, formally, both $n^{f_1(n)}$ and $n^{f_2(n)}$ are in $n^{o(1)}$ class. But $\lim_{n \to \infty} n^{f_1(n)} = \lim_{n \to \infty} \log n = \infty$ and $\lim_{n \to \infty} n^{f_2(n)} = \lim_{n \to \infty} n^{\frac{1}{n}} = 1$ which are quite different cases.
>
> I think notation $O(n \log(n))$ is more established and universally used in the literature, especially if the algorithm is based on FFT (see e. g. [1-3]).
>
> *3.* I agree. Perhaps, the authors could explicitly recite the statement of the corresponding theorem in [4] to make their argument clearer.
>
> *4.* Your construction for this specific case is correct. I think the authors should explicitly include it in the relevant section of the paper. Nevertheless, I don't believe their stronger claim (point 7 in my original review) is true. More on that later.
>
> *6.* Let me politely disagree. The matrix I provided is a symmetric matrix but not a circulant matrix. To see it, note that by definition of a circulant matrix, all rows in it are identical up to a circular shift by some offset.  That's not the case for this matrix. More generally, the definition of $C^{l_1, l_2}$ doesn't imply it is circulant as the elements of $(W_{i-j})_{l_1, l_2} can be arbitrary, unless specifically chosen to induce a circular matrix.
>
> *7.* After having read the latest author comments and the construction in 4, I'm still not convinced pre-exponentiation matrix $M$ could be made circular always by varying $W_{i-j}$ (or $R_{i-j}$, using the paper's notation from another page). It works by specific construction of matrices $Q$, $K$,  $W_{i-j}$ with carefully chosen values. For every fixed set $\\{R_{i-j}\\}$, there exist such values $Q$ and $K$ which do not lead to a circular $M$. Example: for $Q=\begin{bmatrix} 1 \\\\ 2 \end{bmatrix}$ and $K=\begin{bmatrix} 3 \\\\ 4 \end{bmatrix}$, $R_0 \neq 0$ , the result will be
> $\begin{bmatrix}
> 3R_0 & 4R_1 \\\\
> 6R_{-1} & 8R_0
> \end{bmatrix}$ which is not a circular matrix. In practice, $Q$ and $K$ are not fixed and data-dependent while $R_{i-j}$ are constant.
>
> **References**
>
> [1] Fu et al., FlashFFTConv: Efficient Convolutions for Long Sequences with Tensor Cores. ICLR 2024
>
> [2] Gu et al., Efficiently Modeling Long Sequences with Structured State Spaces. ICLR 2022
>
> [3] Lee-Thorp et al., "FNet: Mixing Tokens with Fourier Transforms." NAACL 2022
>
> [4] Josh Alman and Zhao Song. Fast attention requires bounded entries. In NeurIPS, 2023.

---

> > ### Comment · Reviewer_sE6r · 2024-11-28
> > **Response about circulant matrices**
> >
> > Thank you very much to reviewer Q6JF and to the authors for their answers. Let me reply to a couple of points on circulant matrices.
> >
> > 6. My apologies, I went too quickly in my previous comment. I agree with reviewer Q6JF that the matrix is not circulant. The condition that all entries along any diagonal are the same is necessary but not sufficient to get a circulant matrix. One also needs that entries along any two diagonals at distance $n$ are equal, which is not the case in the counter-example of matrix $C^{1,1}$ proposed by reviewer Q6JF. I would be highly interested in the answer of authors on this point, because it questions the applicability of the algorithm for standard RoPE attention.
> > 7. I think that the statement of lines 186-188 applies only to the case $d=1$, in which case $Q$ and $K$ are scalars, so multiplying by $Q$ and $K$ does not change the fact that $M$ is circulant, and the construction proposed by the authors works. I agree with reviewer Q6JF that the construction does not work for $d>1$ (eg with $d=2$ as in the counter-example of the reviewer). In any case, this is in my opinion less problematic than the point above, since the statement we are discussing is a comment of the authors in the introduction, which does not impact the proof.

---

> > > ### Comment · Reviewer_sE6r · 2024-12-02
> > >
> > > Dear authors,
> > >
> > > This is a gentle reminder that there remains unresolved questions raised by reviewer Q6JF (see above). I would be very interested in your answer on those points, in particular point 6. I think this may be an important issue with the paper, so your rebuttal would help us give a fair assessment to the paper.
> > >
> > > Thank you.

---

### Official Review · Reviewer_Tydz · 2024-11-04

**Soundness:** 3
**Presentation:** 3
**Contribution:** 3
**Rating:** 8
**Confidence:** 3

**Summary:**

This paper proposes a method to compute RoPE attention in nearly-linear time.
More formally, given a set of matrices $W_i$, for integer $i \in [-(n-1), (n-1)]$, and matrices $Q$, $K$ and $V$, the matrix $A$ defined as

$$A_{i,j} := \exp\left( Q_{i,\*} W_{i-j} K_{j,\*}^T / d  \right)$$

and $D := diag(A \mathbb{1}_n)$.
The goal is to approximate the matrix $D^{-1} A V$.

This is done through a decomposition of the matrix $A$ into a sum of rescaled circulant matrices $A^{\ell_1, \ell_2}$. This allows to compute a product of $t$ such matrices and a vector in time $O(t n \log n)$ using Fast Fourier Transform. This fact is used to compute the desired matrix using the polynomial method

**Strengths:**

- This paper adapts the algorithm proposed by Alman & Song, (2023; 2024a;b) for a wider variety of attention mechanisms.
This is a fairly important step for the overall optimization of general transformer algorithms.
- The presented method is applicable to a large variety of RoPE-style attention mechanisms.

**Weaknesses:**

The overall structure of the presentation is good. However, there are several minor issues that make the paper difficult to understand. I suggest the following actions:
- In introduction, define what $B$ is before discussing it.
- Replace "We define" in line 410 with "Recall that".
- Move the definition of $A^{\circ t}$ from the beginning of preliminaries inside Lemma 5.1, since it is the only place where it is used.

**Questions:**

All of my questions got resolved during the discussion process. It also allowed me to correctly understand the proofs inside the paper, which altered my recommendation.

---

> ### Author Response · Authors · 2024-11-22
>
> Thank you for the review. The simplification you suggest to claim 4.8 does not work, and touches on the core reason why RoPE attention is more difficult to apply the polynomial method to than regular attention.
>
> If it were true that $A^{\ell_1,\ell_2}$ were an outer product of vectors as you suggest, then it would be a rank 1 matrix, and A, the entire attention matrix before exponentiation, would have low rank at most $|S| \leq O(\log n)$. In fact, the main difficulty with RoPE attention, and the reason this type of approach cannot work, is that $A$ can have full rank even when $d=1$. See the section on page 4 called "Technique Overview: Limitation of Prior Techniques" where we discuss this in more detail.
>
> You're right that $\ell_1, \ell_2$ are the indices into $W_{i-j}$, which is a $d \times d$ matrix. However, in definition 4.7 we are defining an $n \times n$ matrix $C^{\ell_1, \ell_2}$ whose $(i,j)$ entry is equal to the $(\ell_1, \ell_2)$ entry of $W_{i-j}$. Thus, this is a circulant matrix (since its $(i,j)$ entry depends only on $i-j$) but it is an $n \times n$ matrix, and not a scalar as you suggest. On line 412 we relate the $(i,j)$ entry of $A^{\ell_1, \ell_2}$ to the $(i,j)$ entry of $C^{\ell_1, \ell_2}$, from which the matrix relation we give on line 415 follows. Since $C^{\ell_1, \ell_2}$ is an $n \times n$ circulant matrix, this does not show that $A^{\ell_1, \ell_2}$ has rank 1 as you are suggesting.
>
> Finally, to respond to the specific bullet points you raise in the weaknesses section:
> - $W_{i-j}$ is the matrix given as input to the problem in Definition 1.1; this refers to the $(\ell_1, \ell_2)$ entry of that matrix.
> - As above, such a simplification is not possible.
> - Thanks for noticing this, we will define $B$ earlier.
> - In Claim 4.8 we wanted to remind the reader the definition of $C^{\ell_1, \ell_2}$ to emphasize that we are about to use it.
> - We do use this notation, for instance on line 450 on page 9.

---

> > ### Comment · Reviewer_Tydz · 2024-11-24
> >
> > Thank you for clarification! I've updated the review accordingly.

---

> > > ### Author Response · Authors · 2024-11-26
> > > **Thank you!**
> > >
> > > We are glad our response addressed your concerns. We appreciate your efforts and time. Thank you for your helpful suggestions and score raising!

---

### Meta-Review · Area_Chair_cZdV · 2024-12-20

**Metareview:**

This paper proposes a method to allow for the previously developed methods to compute trainsformer operations more efficiently to be extended to the setting where RoPE positional embeddings are used in the network.

Several reviewers found the paper novel and interesting, but unfortunately other reviewers noted potential issues with both the clarity and correctness of the work. In particular, there appears to be a potentially serious technical issue with whether matrices which are required to be circulant for fast computation via FFT will actually be circulant when the method is applied.  One reviewer provided a specific example matrix of the issue, which the authors claim is circulant in the discussion, but I am in agreement with the reviewer that the provided example matrix is Topelitz but not circulant.  Additionally, the reviewers also note that there are no experiments showing the computational improvement of the method.

Overall, given the highly technical nature of the work, it would seem that the work would be better served by improving the clarity of the presentation to convince the reader of the technical correctness of the method, along with potentially addressing some of the technical correctness issues raised by the reviewers and providing an experimental demonstration of the benefits of the method.

**Additional Comments On Reviewer Discussion:**

There was a lengthy discussion both between the reviewers and the authors as well as between the reviewers.  Overall it was concluded that some of the technical correctness issues raised in some of the reviewers were not yet sufficiently address.

---

### Decision · Program_Chairs · 2025-01-22

Reject